# C-Terminal Domain of the Human Zinc Transporter hZnT8 Is Structurally Indistinguishable from Its Disease Risk Variant (R325W)

**DOI:** 10.3390/ijms21030926

**Published:** 2020-01-31

**Authors:** Raheem Ullah, Aamir Shehzad, Majid Ali Shah, Matteo De March, Fouzia Ismat, Mazhar Iqbal, Silvia Onesti, Moazur Rahman, Michael J. McPherson

**Affiliations:** 1Drug Discovery and Structural Biology group, Health Biotechnology Division, National Institute for Biotechnology and Genetic Engineering (NIBGE), Faisalabad 44000, Pakistan; raheem.biotech@gmail.com (R.U.); aamiruopbbt@yahoo.com (A.S.); majidshah1@yahoo.com (M.A.S.); fouziaismat@yahoo.com (F.I.); hamzamgondal@gmail.com (M.I.); 2Structural Biology Laboratory, Elettra-Sincrotrone Trieste, 34149 Trieste, Italy; matteo.demarch@elettra.eu (M.D.M.); silvia.onesti@elettra.eu (S.O.); 3Astbury Centre for Structural Molecular Biology and School of Molecular and Cellular Biology, Faculty of Biological Sciences, University of Leeds, Leeds LS2 9JT, UK

**Keywords:** human zinc transporter hZnT8, C-terminal domain, disease risk variant, biophysical characterization

## Abstract

The human zinc transporter 8 (hZnT8) plays important roles in the storage of insulin in the secretory vesicles of pancreatic β cells. hZnT8 consists of a transmembrane domain, with its N- and C-termini protruding into the cytoplasm. Interestingly, the exchange of arginine to tryptophan at position 325 in the C-terminal domain (CTD) increases the risk of developing type 2 diabetes mellitus (T2D). In the present study, the CTDs of hZnT8 (the wild-type (WT) and its disease risk variant (R325W)) were expressed, purified, and characterized in their native forms by biophysical techniques. The data reveal that the CTDs form tetramers which are stabilized by zinc binding, and exhibit negligible differences in their secondary structure content and zinc-binding affinities in solution. These findings provide the basis for conducting further structural studies aimed at unravelling the molecular mechanism underlying the increased susceptibility to develop T2D, which is modulated by the disease risk variant.

## 1. Introduction

Zinc (Zn^+2^) plays important roles in cellular growth and differentiation [1,2,3,4]. A large number of cellular proteins bind Zn^+2^, and many others are involved in its transport [5,6]. In humans, cellular Zn^+2^ homeostasis is maintained by two families of Zn^+2^ transporters—the cation diffusion facilitator (CDF) family, also known as the SLC30 family, that controls the Zn^+2^ concentration in the extracellular matrix or intracellular vesicles, and the Zrt- and Irt-like proteins (ZIP) family, also known as the SLC39A family, that is implicated in controlling the intracellular Zn^+2^ uptake [6,7,8]. The CDF family consists of ten (10) Zn^+2^ transporters (ZnTs), named ZnT1 (or SLC30A1) to ZnT10 (or SLC30A10), which facilitate Zn^+2^ transport across cellular membranes [6,7,8]. ZnT and ZIP-family members play key roles in various biological functions, and an understanding of their importance in health and disease continues to increase as more information on disease-related mutations in the corresponding protein-encoding genes comes to light [9]. Variations in the amino acid sequence of ZnT and ZIP-family members are associated with various diseases such as type 1 diabetes mellitus (T1D), type 2 diabetes mellitus (T2D), Parkinsonism, hepatic cirrhosis, hypermanganesemia, dystonia, polycythemia, acrodermatitis enteropathica, and nonsymptomatic high myopia [9].

In humans, the human zinc transporter 8 (hZnT8) is implicated in decreasing the cytoplasmic Zn^+2^ level by exporting Zn^+2^ ions into insulin-containing secretory vesicles in pancreatic β cells [10,11]. Notably, hZnT8 is involved in the formation of water-insoluble insulin crystals, ultimately resulting in the storage of the hexameric insulin bound to two Zn^+2^ ions in pancreatic β cells [12,13]. Hence, hZnT8 has been regarded as a metal carrier during insulin maturation, storage, and secretion [10]. It has been shown that a nonsynonymous single nucleotide polymorphism (SNP), termed as rs13266634, in the gene (*SLC30A8*) encoding hZnT8 introduces a substitution (R325W) in the C-terminal domain (CTD) of hZnT8 (Figure 1), and genome-wide association studies (GWASs) reveal that the substitution R325W in the CTD of hZnT8 increases the risk of developing T2D [14]. However, the link between the rs13266634 polymorphism and the risk of susceptibility to T2D has not been identified in all of the human studies [15,16,17,18,19]. Interestingly, it has been shown that loss-of-function mutations in the gene encoding hZnT8 provide protection against the risk of T2D [20]. Such conflicting study results may be due to racial differences of the selected population, different criteria for sample selection, or alternative assay systems used for detection [15].

The CTD of hZnT8 plays important roles in protein–protein interactions and Zn^+2^ sensing [15,21,22,23,24]. Modeling studies based on bacterial homologues have predicted that the residue at position 325 in hZnT8 is shielded from the solvent by the CTD β strands, suggesting that the Zn^+2^ sensing capacity of the disease risk variant remains unaffected [25]. However, a recent study has shown that bacterial models are not suitable for explaining the structure and function of hZnT8 owing to their different physiological roles in bacterial cells [24]. As hZnT8 is a transmembrane protein, it is highly challenging to purify the protein in its native form for conducting functional studies [15]. Recently, a proteoliposome-based full-length hZnT8 was produced through a combined bioengineering and nanotechnology approach [26]. Hence, most of the studies have focused only on the soluble CTD instead of the full-length protein in order to investigate hZnT8 activity [15,24]. As protocols for the expression and purification of the hZnT8 CTDs in their native forms have not been reported so far, data on the biophysical characterization of the CTDs in their native forms are lacking in the literature. In this report, we present results obtained from the biophysical characterization of the native form of the hZnT8 CTD and its disease risk variant in solution. These proteins were expressed and purified using a modified strategy that resulted in better solubility of the CTDs, allowing us to characterize the CTDs in their native forms in solution. The data indicate that the CTDs form tetramers, and exhibit negligible differences in their secondary structure content and Zn^+2^-binding affinities. Thus, this study could provide the basis for conducting further structural investigations of hZnT8 in the future in order to unravel molecular mechanisms underlying the increased susceptibility to develop various diseases, which are modulated by disease risk variants. Moreover, the characterization of the native form of the CTDs in the current study clarifies that the R325W substitution is not much associated with T2D.

## 2. Results

### 2.1. The Maltose-Binding Protein (MBP) Tag Improves the Solubility of the CTDs

A significant amount (60 mg per 1 L culture of the cells) of soluble proteins was obtained from the expression of the His-MBP-TEV-fused CTDs in *Escherichia coli* BL21 DE3 (Appendix A). Both proteins were successfully purified in their native forms using chromatographic techniques. As the His tag has affinity for divalent metal ions, and therefore may affect the Zn^+2^ binding analysis, the N-terminal tags were removed from the CTDs after the nickel-nitrilotriacetic acid (Ni-NTA) affinity chromatography (Appendix A). The data presented here suggest that the introduction of the MBP tag between the His tag and the TEV site improves the solubility of the CTDs during expression and purification steps (Appendix A).

### 2.2. The CTDs Exist as Tetramers in Solution

Monodispersity analysis by negative-stain electron microscopy (EM) revealed that both proteins were devoid of any aggregates in solution (Appendix A). In order to determine the oligomeric state, size-exclusion chromatography (SEC) of the wild-type (WT) CTD was performed under native conditions. The chromatogram of the WT CTD showed a major peak with a retention time corresponding to the apparent molecular mass of 40 kDa, suggesting that the protein exists as a tetramer in solution (Figure 2a). The latter findings were also supported by analytical ultracentrifugation using the sedimentation velocity method, revealing a major peak corresponding to the tetrameric form (39.6 kDa) for the WT CTD (Figure 2c). Moreover, the experimental molecular weight (MW) of the proteins was also confirmed using mass spectrometry, revealing that the MW of the WT CTD is 9.828 kDa (Appendix A). The SEC elution profile of the WT CTD under reducing conditions (3 mM beta-mercaptoethanol in 20 mM Tris-Cl, pH 7.5, 150 mM NaCl) was similar to that under native conditions, suggesting that the tetrameric state is not due to disulfide linkages (Figure 2d). Interestingly, the equilibrium shifted from the tetrameric state to the monomeric state in the presence of 5 mM ethylenediaminetetraacetic acid (EDTA) (Figure 2d), indicating that Zn^+2^ ions play an important role in stabilizing the tetrameric state of the CTD. This tetrameric state is also stable under denaturing conditions (8 M urea), although some aggregates are also formed (Figure 2d). Importantly, the CTD of the disease risk variant also exists in the tetrameric form similar to the WT CTD (Figure 2b), suggesting that the R325W substitution does not affect the oligomeric state of the CTD in solution.

### 2.3. The CTDs Bind Zn^+2^ with High Affinity

To assess the Zn^+2^-binding affinity of the CTDs, we exploited the intrinsic fluorescence of proteins. Initial experiments were performed to observe any detectable change in intrinsic fluorescence upon Zn^+2^ addition. For this, emission spectra were recorded in the absence and presence of 50 mM Zn^+2^. Addition of saturating concentrations of Zn^+2^ to samples of the CTDs induced a significant quenching (~9%) (Figure 3). Similar experiments with the buffer alone did not result in significant quenching. The relationship between the fluorescence quenching for the CTDs and the concentration of Zn^+2^ used is depicted in Figure 3. The quenching increased upon Zn^+2^ addition with an approximately hyperbolic trend showing saturation at high Zn^+2^ concentrations.

The best fit of the binding curve data was obtained using a two-binding-site model as described in Materials and Methods (4.6. Metal Binding Analysis). For the WT CTD, *K_d_* values of 0.03 ± 0.01 and 0.81 ± 1.60 µM were obtained, whereas *K_d_* values of 0.04 ± 0.02 and 2.54 ± 4.72 µM were obtained for the R325W CTD. These results reveal that both the variants bind Zn^+2^ with high affinity, suggesting that the R325W substitution is compensated in the onset of T2D.

### 2.4. The CTDs Share Similar Structural Features in the Presence and Absence of Zn^+2^

In order to investigate whether the CTDs display similar structural features, circular dichroism (CD) spectroscopy of the proteins was performed. The CD spectrum of the WT CTD, particularly in the region between 200 and 240 nm that reflects the secondary structure propensities in proteins [27], is almost identical to the CTD of the disease risk variant (Figure 4). Thus, it appears that the R325W substitution does not induce significant structural changes in the CTD. Moreover, addition of 1 mM Zn^+2^ to the CTDs did not produce any detectable structural changes in solution, indicating that Zn^+2^ binding is independent of any alteration in the global structure of the CTDs (Figure 4). Finally, to check the thermal and refolding ability of the CTDs, CD spectra were recorded for the proteins after heating (unfolding) and cooling (refolding). These spectra showed that the domains could not refold after thermal unfolding (Figure 4).

To estimate the secondary structure content of the CTDs, Fourier transform infrared (FT-IR) spectroscopy of the proteins was performed, revealing that the CTDs fold into mixed secondary structure components in solution (Figure 5). To provide a quantitative estimate of the α-helical and β-sheet content of the WT CTD and the R325W CTD, the experimental spectra on the amide I region were reproduced by a fit of seven and nine components, respectively, dominated by those indicative of α-helices and β-sheets [28,29]. Our quantitative analysis predicted the presence of ~27% to 29% α-helices and ~24% to 26% β-sheets for the CTDs (Figure 5 and Table 1), suggesting that the R325W substitution does not cause any significant structural changes in the CTD of hZnT8 in solution.

## 3. Discussion

In the current study, we compared the biophysical properties of the CTD of hZnT8 with its disease risk variant (R325W). The CTDs were successfully expressed in *E. coli* and purified to homogeneity in the native form by chromatographic techniques. Gel filtration analysis reveals that the CTDs predominantly exist in tetrameric forms in solution. Surprisingly, the tetramers are also stable in the presence of denaturing concentrations (8 M) of urea. This oligomeric state is stabilized by Zn^+2^, as the addition of a chelating agent, such as EDTA, results in the dissociation of homotetramers into monomers. To the best of our knowledge, the tetrameric form of the CTD of ZnT8 has not been reported previously. In mouse pancreatic cell lines, the endogenous ZnT8 protein exists as homodimers that require membrane components for dimerization [34]. Recently, it has been reported that the CTDs of hZnT8 exist as dimers in solution [24]. However, the CTDs of hZnT8 were characterized with uncleaved N-terminal tags as the proteins were unstable and precipitated when the tags were removed [24]. In the present study, the tags were cleaved during purification steps, and the resultant native variants were stable and no precipitation was observed. However, it is not clear whether the oligomeric state of the CTDs of hZnT8 is influenced by the presence of N-terminal tags.

The intrinsic fluorescence exhibited by the CTDs of hZnT8 was exploited to confirm their affinity for binding Zn^+2^. The data presented here indicate that Zn^+2^ ions associate with the CTD of hZnT8 and its disease risk variant with high affinity. In a previous study, it was reported that the CTD of hZnT8 has at least two Zn^+2^-binding sites [24], explaining why the domain binds Zn^+2^ with high affinity. No significant difference in the binding affinity for Zn^+2^ is observed between the CTDs, suggesting that the Zn^+2^-binding capacity of the R325W CTD remains unaffected, which might be due to the location of the substitution in the protein structure. In the crystal structure of a homologous zinc transporter, YiiP in *E. coli*, the residue at position 325 is shielded by the planar surface of three cytoplasmic β-sheets [21,25].

Currently, the three-dimensional structure of the full-length hZnT8 or the CTD is not available in the Protein Data Bank (PDB). In order to estimate the secondary structure content, biophysical characterization of the CTDs of hZnT8 (WT and its R325W variant) was performed through CD and FT-IR spectroscopic techniques. The data do not reveal any structural difference between both the CTDs when analyzed using a given technique (Table 1). Differences in the overall quantitative estimation of the secondary structure content obtained by CD and FT-IR spectroscopic techniques could be due to differences in the techniques employed and algorithms used for the analysis of the secondary structure. Importantly, the addition of 1 mM Zn^+2^ does not cause any noticeable structural change in the CTDs. The latter findings suggest that the R325W substitution in hZnT8 is tolerated and results in the transport of reasonable amounts of Zn^+2^ to the correct sites in pancreatic β cells, corroborating previous reports [25]. However, our study does not rule out the possibility that local structural changes, not detectable through CD and FT-IR techniques, in the CTDs could occur that would increase the risk of developing T2D in the case of the disease risk variant. Efforts are underway to crystallize the hZnT8 CTD and its disease risk variant for solving high-resolution crystal structures of the CTDs through X-ray crystallography in order to observe local structural changes in the CTDs.

## 4. Materials and Methods 

### 4.1. Cloning, Expression, and Purification of the CTDs

DNA encoding the CTD of hZnT8 (Ser281–Asp369) was amplified from the cDNA (IRCMp5012C0519D; pICMV) by “touchdown” PCR [35] using primers (5′ GCAGCCATATGAAGAGCCTGAATTACAGTGGTGTG 3′ and 5′ GCTCGAATTCTTAGTCACAGGGGTCTTCACAG 3′). The PCR reaction mixture consisted of 1 μL of template DNA, 0.2 mM dNTPs, 1.5 mM MgSO_4_, 0.5 μM of each primer, 1.25 U of Pfu DNA polymerase and 1X Pfu buffer (200 mM Tris-HCl, pH 8.8, 100 mM (NH_4_)_2_SO_4_, 100 mM KCl, 1% Triton X-100, 1 mg/mL BSA) (Fermentas, St. Leon-Rot, Germany). The amplified product was cloned into a pET28a vector using molecular biology methods. Using this recombinant plasmid as a template, the construct for the R325W variant of hZnT8 was generated by the Stratagene QuikChange site-directed mutagenesis kit. The resultant vectors allowed the expression of the CTDs fused to a His tag, an MBP tag, and the TEV site at the N-terminus of the target proteins. The constructs were first introduced into *E. coli* OmniMAX^TM^ 2T1 strains and then into *E. coli* BL21 (DE3) expression host cells by a chemical transformation method. Transformed *E. coli* BL21 (DE3) cells were grown in LB shaking flask cultures supplemented with 25 µg/mL of kanamycin at 37 °C. When the OD_600_ reached 0.6, the expression of the CTDs was induced by adding 1 mM isopropyl β-d-1-thiogalactopyranoside (IPTG) into the broth cultures. The cells were further grown at 22 °C for 5 h in shaking flask cultures, upon which the cells were harvested through centrifugation. The cell pellet was re-suspended in buffer (20 mM Tris-Cl, pH 7.5, 300 mM NaCl, 5 mM imidazole, 4 mM β-mercaptoethanol) and lysed by two passes through a cell disruptor (Constant Systems Ltd., Daventry, Northants, UK). The cell extract was centrifuged at 12,000× *g* for 25 min at 4 °C to remove cell debris. The supernatant was passed through Ni-NTA agarose (Qiagen, Hilden, Germany) resin pre-equilibrated with buffer (20 mM Tris-Cl, pH 7.5, 300 mM NaCl, 5 mM imidazole, 4 mM β-mercaptoethanol). Subsequently, the column was washed with 30-column volumes of buffer (20 mM Tris-Cl, pH 7.5, 300 mM NaCl, 30 mM imidazole, 4 mM β-mercaptoethanol). The CTDs were eluted with buffer (20 mM Tris-Cl, pH 7.5, 150 mM NaCl, 150 mM imidazole, 4 mM β-mercaptoethanol). To produce the CTDs in native form, the fused tags at the N-terminus were removed by incubating the CTDs with oligohistidine-tagged tobacco etch virus protease in buffer (20 mM Tris-Cl, pH 7.5, 100 mM NaCl, 10% glycerol, 2 mM β-mercaptoethanol, and 1 mM EDTA) at 25 °C using 10:1 protein-to-protease ratio. The mixture was dialyzed overnight against buffer (20 mM Tris-Cl, pH 7.5, 10 mM NaCl), and then subjected to anion-exchange chromatography using Hi-Trap Q column (GE Healthcare Life Sciences, Uppsala, Sweden). The column was equilibrated with buffer (20 mM Tris-Cl, pH 7.5, 10 mM NaCl). The CTDs eluted when the salt concentration ranged from 50 to 700 mM. The fractions containing the CTDs were pooled and passed 3 or 4 times through Ni-NTA affinity columns to remove oligohistidine-tagged proteins using a fresh Ni-NTA affinity resin each time. In order to remove any remaining MBP-tagged proteins, purified CTDs were passed through amylose resin (New England Biolabs, Ipswich, MA, USA) pre-equilibrated with buffer (20 mM Tris-Cl, pH 7.5, 150 mM NaCl, 1 mM β-mercaptoethanol).

The protein concentration was determined using a NanoDrop 2000c spectrophotometer (Thermo Fisher Scientific, Wilmington, DE, USA) using calculated extinction coefficients of 6990 and 12,490 M^−1^ cm^−1^ for the WT CTD and the R325W CTD, respectively, at 280 nm. The CTDs were concentrated to 10–20 mg/mL in buffer (20 mM HEPES, pH 7.5, 150 mM NaCl) using Amicon Ultra centrifugal filters (Millipore, Carrigtwohill, County Cork, Ireland) with a 5 kDa molecular weight cut-off.

### 4.2. Gel Filtration

The purified CTDs (concentration 10 mg/mL) were analyzed by SEC using an ÄKTA-pure chromatography system (Superdex 75 column - GE Healthcare Life Sciences). The running buffer (20 mM Tris, pH 7.5, 150 mM NaCl) was supplemented with either a denaturant (3 mM β-mercaptoethanol or 8 M urea) or a chelating agent (5 mM EDTA) as required. Commercially available standard proteins (aldolase, ovalbumin, carbonic anhydrase, and ribonuclease; GE Healthcare) were used for calibration.

### 4.3. Analytical Ultracentrifugation

The oligomeric state of the WT CTD (6 mg/mL in buffer (20 mM Tris, pH 7.5, 150 mM NaCl)) was further analyzed by sedimentation velocity ultracentrifugation [36] using a Beckman ultracentrifuge (Beckman Coulter Inc., Indianapolis, IN, USA). The data were analyzed using the program SEDFIT (version 14.1, National Institutes of Health, Bethesda, MD, USA) [36] to calculate the sedimentation coefficient distribution of the WT CTD.

### 4.4. Mass Spectrometry

The identity of the CTDs was confirmed by mass spectrometry at the Taplin Mass Spectrometry facility, Harvard Medical School, Boston (http://taplin.med.harvard.edu/), by peptide mass fingerprinting using chymotrypsin that cleaves at C-terminal to the amino acid sequence (FYWML), not before prolines, under low specificity.

### 4.5. Negative-Stain EM

The CTDs were analyzed by negative-stain EM, as described previously [37]. A drop (3 to 4 µL) of protein solution was applied to a glow-discharged carbon-coated copper–palladium grid and incubated at room temperature for 1 min. The grid was washed with ddH_2_0, incubated with 0.75% uranyl formate, blotted, and dried.

### 4.6. Metal Binding Analysis

Fluorescence measurements were performed at 25 °C on a scanning fluorimeter (PerkinElmer, Shelton, CT, USA) using 3 mL quartz cuvettes. Excitation and emission slit widths of 2 nm were used. Solutions (2.5 mL) of purified proteins in buffer (20 mM HEPES, pH 7.5, 150 mM NaCl) were serially prepared by adding ZnCl_2_ to a final concentration of 50 mM. Excitation was done at 285 nm and emission spectra were recorded from 310 nm to 425 nm. The affinity constant (*K_d_*) for Zn^+2^ was calculated from the background-subtracted spectra by averaging three measurements and using a two-binding-site model to fit the data as follows:(1)$$∆F\;=\frac{\left(∆\mathrm{Fmax}1\;\times X\right)}{\left(Kd1\;+X\right)}+\;\frac{\left(∆\mathrm{Fmax}2\;\times X\right)}{\left(Kd2\;+X\right)},$$
where ∆F is the change in fluorescence for a given concentration of substrate X, and ∆Fmax1 and ∆Fmax2 are the total fluorescence changes for the first and second Zn^+2^-binding sites, respectively. *K_d_*1 and *K_d_*2 are dissociation constants for the first and second Zn^+2^-binding sites, respectively.

### 4.7. CD Spectroscopy

CD spectroscopic measurements were carried out using a J-810 spectropolarimeter (JASCO, Easton, MD, USA) with Hellma quartz cuvettes at 25 °C and with a constant nitrogen flush. Samples of purified proteins were dialyzed in buffer (10 mM potassium phosphate, pH 7.4) and concentrated to 0.3 mg/mL. Spectra (185–260 nm) were then recorded as an average of 10 accumulations, with the background of the buffer subtracted. Spectra were recorded at a speed of 1 nm/15 s, sensitivity 50 mdeg, bandwidth 1 nm, resolution 1 nm, and response time 15 s. The samples were scanned at 25 °C after manually heating to 90 °C and then cooling to 25 °C for 5–10 min.

### 4.8. FT-IR Spectroscopy

The CTDs (20 µL each; 6 mg/mL in buffer (20 mM HEPES, pH 7.5, 150 mM NaCl)) were dried on a platinum ATR platform (diamond crystal) to form a hydrated film. Spectra from 4000 to 500 cm^−1^ were acquired on a Bruker FT-IR spectrometer (Billerica, MA, USA) in absorbance mode at 1 nm resolution, 256 scans co-addition and three-term Blackman–Harris apodization. To remove the bending vibration of H_2_O that gives an absorption band at around 1645 cm^−1^ in the amide I region, spectra for the buffer were recorded, and subtraction of residual vapor absorption was performed. Final manipulated spectra of both domains were obtained between 2000 cm^−1^ and 1770 cm^−1^. For secondary structure analysis, the second derivative of spectra corresponding to the amide I region (1700–1600 cm^−1^) was obtained using Origin software version 7.0 (Origin Lab Corporation, Northampton, MA, USA) following the Savitzky–Golay method [28]. Decomposition of each spectrum into individual bands in the amide I region was performed by nonlinear peak fitting using Galactic PeakSolve™ software (version 1.05, Galactic Industries Corporation, Salem, NH, USA). Band assignments for the interpretation of spectra were done on the basis of previous measurements [29]. 

## 5. Conclusions

In the present study, we successfully expressed and purified the CTDs of hZnT8 (the WT and the R325W variant) in their native forms. Unlike many other ZnTs, the CTDs exist as tetramers, and bind Zn^+2^ with high affinity. Structural characterization of the variants in solution reveals that the R325W substitution does not cause any significant structural perturbations in the CTD of hZnT8. Thus, this study could provide the basis for conducting further structural investigations of the CTDs in their native forms through X-ray crystallography or small-angle X-ray scattering (SAXS), for example, in the future.

## Figures and Tables

**Figure 1 ijms-21-00926-f001:**
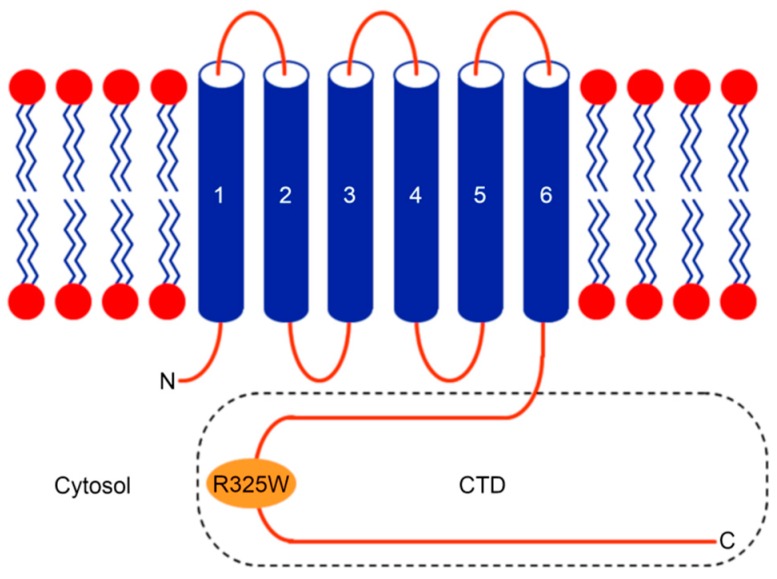
Cartoon diagram of the human zinc transporter 8 (hZnT8) protein topology depicting the location of the R325W substitution in the C-terminal domain (CTD). Like many other Zn^+2^ transporters (ZnTs), hZnT8 consists of a transmembrane domain (TMD), with its N- and C-termini facing the cytoplasm [8].

**Figure 2 ijms-21-00926-f002:**
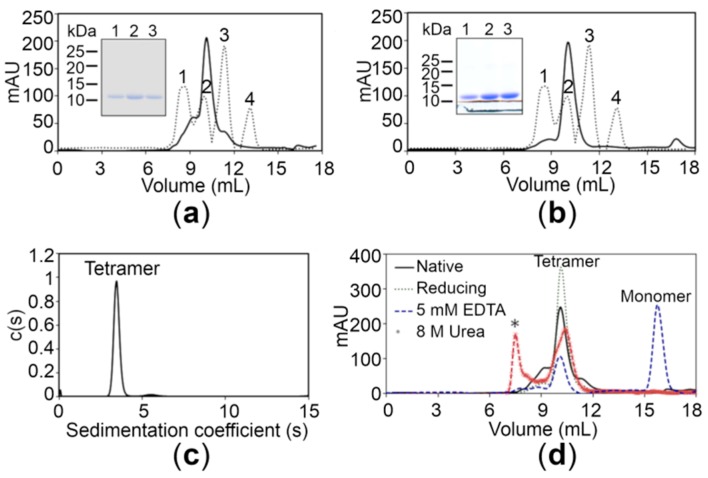
Analysis of the hZnT8 CTDs through gel filtration and analytical ultracentrifugation. (**a**,**b**) Gel filtration chromatograms of the wild-type (WT) CTD (**a**) and the R325W CTD (**b**) showing an apparent molecular mass of approximately 40 kDa. The dotted lines represent the following protein standards: aldolase (158 kDa) (peak 1), ovalbumin (44 kDa) (peak 2), carbonic anhydrase (29 kDa) (peak 3), and ribonuclease (13.7 kDa) (peak 4). Inset shows the SDS-PAGE analysis of the WT CTD (**a**) and the R325W CTD (**b**) purified by size exclusion chromatography. (**c**) A major peak corresponding to the tetrameric form (39.6 kDa) for the WT CTD obtained using analytical ultracentrifugation. (**d**) Gel filtration chromatograms of the WT CTD with 5 mM ethylenediaminetetraacetic acid (EDTA) (blue dotted lines), 8 M urea (brown dotted lines), and 3 mM β-mercaptoethanol (green dotted lines) are depicted. Protein aggregates formed in the presence of urea are shown with an asterisk (*).

**Figure 3 ijms-21-00926-f003:**
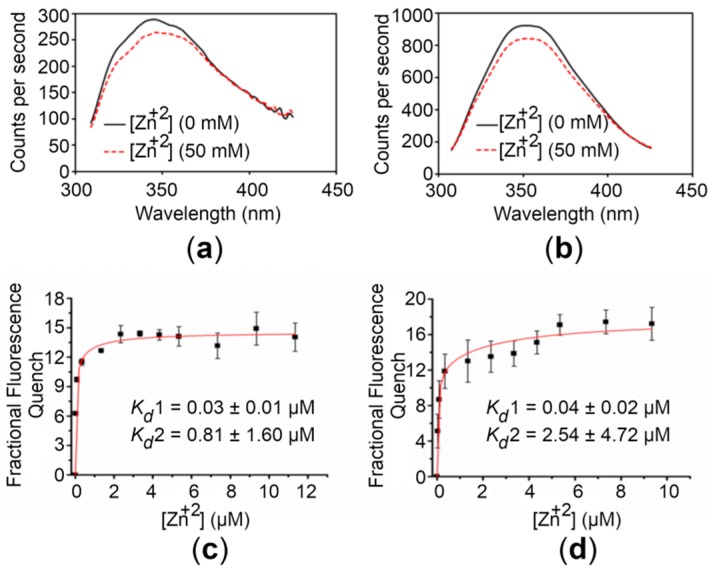
Zn^+2^ binding analysis of the hZnT8 CTDs. (**a**,**b**) Fluorescence emission spectra of the WT CTD (**a**) and the R325W CTD (**b**) before (continuous lines) and after (dotted lines) addition of 50 mM Zn^+2^ are shown. (**c**,**d**) Concentration-dependent Zn^+2^-binding affinity curves for the WT CTD (**c**) and the R325W CTD (**d**). The apparent affinities (*K_d_* values) for Zn^+2^ binding to the WT CTD and the R325W CTD were obtained by nonlinear fitting of the respective quench/concentration curves to a two-binding-site model. The excitation and emission wavelengths used in the experiments were 285 nm and 350 nm, respectively. Fluorescence was recorded at excitation and emission slit widths of 2.0. Data points and error bars represent the mean and standard deviation of three individual measurements, respectively.

**Figure 4 ijms-21-00926-f004:**
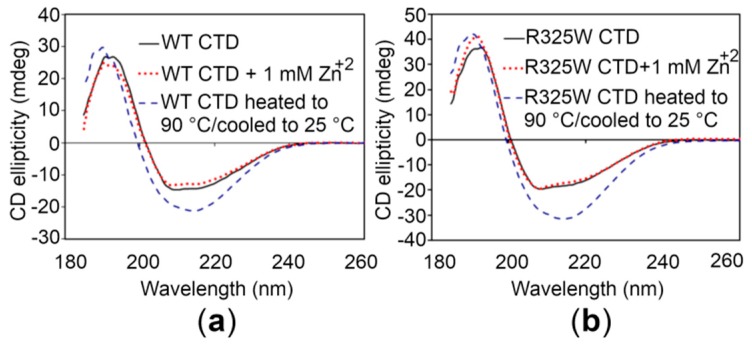
Circular dichroism (CD) spectroscopy of the hZnT8 CTDs. (**a**,**b**) CD spectra of the WT CTD (**a**) and the R325W CTD (**b**) in the absence (black continuous lines) and in the presence (red dotted lines) of 1 mM Zn^+2^. Blue dotted lines represent the CD spectra of the CTDs that were heated up to 90 °C followed by an attempt of refolding upon cooling to 25 °C.

**Figure 5 ijms-21-00926-f005:**
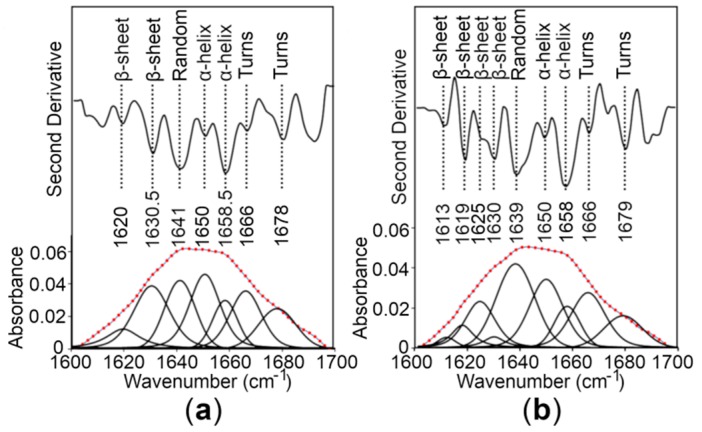
Analysis of the secondary structure content of the hZnT8 CTDs by Fourier transform infrared (FT-IR) spectroscopy. (**a**,**b**) The second derivative and amide I region of FT-IR spectra of a hydrated film of the WT CTD (**a**) and the R325W CTD (**b**) are shown as thick grey lines. The bands obtained by deconvolution are also depicted. The dotted red line represents the curve fitted using the component bands. The peak position of the spectral components and their assignments are also shown.

**Table 1 ijms-21-00926-t001:** Secondary structure content of the WT CTD and the R325W CTD analyzed through spectroscopic techniques in the current study.

Protein	α-Helix (%)	β-Sheet (%)	β-Turn (%)	Random Coil (%)	Technique
WT CTD	41	15	13	31	CD spectroscopy ^1^
	46	10	17	27	CD spectroscopy ^2^
R325W CTD	46	12	13	29	CD spectroscopy ^1^
	53	10	17	20	CD spectroscopy ^2^
WT CTD	29.6	26.1	27	17.3	FT-IR spectroscopy
R325W CTD	27	24	25	24	FT-IR spectroscopy

^1^ CD spectra of the WT CTD and the R325W CTD in the absence of Zn^+2^ (shown in Figure 4; black continuous lines) were analyzed with the protein concentration-independent method using a web-based server at http://perry.freeshell.org/raussens.html [30]. ^2^ CD spectra were analyzed using the DichroWeb server at http://dichroweb.cryst.bbk.ac.uk/html/home.shtml [31,32] and the program CONTIN/reference set 3 [33].

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
