# Peer review of "C-Terminal Domain of the Human Zinc Transporter hZnT8 Is Structurally Indistinguishable from Its Disease Risk Variant (R325W)"

_ijms, 2020, doi:10.3390/ijms21030926_

Round 1
Reviewer 1 Report
The work by McPherson et al. describes a biophysical characterization of human Zn transporter and a disease risk variant (R235W). The authors conclude that both species are oligomeric and they show the similar structural features. The work is well-written but, in my view, there are some serious major flaws that need to be addressed before publication (and they will require more experiments). A point-by-point analysis follows: (a) The sample contains Zn (as concluded from the experiments in SEC with EDTA), then, how the titration experiments were carried out? This reader presumes that, unless the sample was previously treated , the both species used in the titration could contain Zn, and then the results described in Fig. 2 are not reliable. The authors fit the titration data to a one-single site, but in the Discussion (line 221) they say that the CTD has two Zn-binding sites, then what is the significance of the reported association constants. The authors should use other equations to fit the data. I am concerned about the solubility product of Zn(OH)2 (14.8) and the possible formation of complexes with OH- at the pH 7.2, where titration experiments were carried out (line 303). At that pH all those species are formed at the Zn concentrations used, the authors must take into account those other possible equilibria. The authors should use ITC to determine reliable affinity constants for both protein species. (b) The authors use the prediction of structure from CD from other authors (Table 1, reference 54): why did they not use their own spectra (Fig. 3) and the Dichroweb page?. In that way, the comparison in Table 1 will be more reliable. Moreover, other elements of structure prediction (i.e. random coil and beta-turns) should be included in that table (these other types of “structure” are also predicted (deconvoluted) by FTIR). (c) Why did the authors not estimate the presence of self-association equilibria from equilibrium ultracentrifugation, instead of using sedimentation velocity (lines 290-295)? (d) Further experimental details are required for the CD experiments (i.e., response, scan speed, bandwidth). (e) I think the legend of Fig. S1 is wrong (there are two lanes 4 with different meaning).Author Response
Response to Reviewer 1 Comments
We thank the reviewer for valuable suggestions and comments which were very fruitful in improving the quality of the manuscript. Please find below a point-by-point response (highlighted in red) to the reviewer’s comments. The text from the manuscript is flanked by quotation marks. Reference numbers are as per order given in the main text.
Point 1: The work by McPherson et al. describes a biophysical characterization of human Zn transporter and a disease risk variant (R235W). The authors conclude that both species are oligomeric and they show the similar structural features. The work is well-written but, in my view, there are some serious major flaws that need to be addressed before publication (and they will require more experiments). A point-by-point analysis follows: (a) The sample contains Zn (as concluded from the experiments in SEC with EDTA), then, how the titration experiments were carried out? This reader presumes that, unless the sample was previously treated, the both species used in the titration could contain Zn, and then the results described in Fig. 2 (Figure 3 in revised version) are not reliable. The authors fit the titration data to a one-single site, but in the Discussion (line 221) they say that the CTD has two Zn-binding sites, then what is the significance of the reported association constants. The authors should use other equations to fit the data. I am concerned about the solubility product of Zn(OH)2 (14.8) and the possible formation of complexes with OH- at the pH 7.2, where titration experiments were carried out (line 303). At that pH all those species are formed at the Zn concentrations used, the authors must take into account those other possible equilibria. The authors should use ITC to determine reliable affinity constants for both protein species.
Response 1: The main objective of performing the zinc (Zn+2) binding analysis was to investigate the comparative binding behaviour of the wild-type C-terminal domain (WT CTD) and the variant (R325W CTD) for Zn+2 under similar conditions, and to show that the CTDs are functional (i.e. actively bind Zn+2).
We have observed during SEC experiments that the oligermic form of the CTDs is converted into the monomeric form at 5 mM EDTA concentration (Figure 2). In order to remove Zn+2 from the CTDs for titration experiments, we have used 1 mM EDTA in the cleavage buffer while cleaving oligohistidine tags from the CTDs by the TEV protease, assuming that essential Zn+2 ions required for oligomerization of the CTDs would not be removed (as the monomeric form may not be the true physiological state of the CTDs). The CTDs purified after treatment with 1 mM EDTA (as used in a previous study [33]) were subjected to Zn+2 binding analysis.
In this comparative Zn+2 binding study, we previously used a one binding site model to fit the titration data. As per suggestions of both reviewers, we have now used a two binding site model to fit the data (see Materials and methods) which produces a nice curve fitting, suggesting that there exist two Zn+2 binding sites in the CTDs (Figure 3) in agreement with previous report [33]. We are highly grateful to expert reviewers for this major correction and improvement of our manuscript.
Regarding the worthy reviewer's comment about the solubility of Zn+2 at higher pH values, we are aware that Zn+2 ions can make complexes with OH- and become insoluble at higher pH, especially when Zn+2 is salted with SO4-2 (ZnSO4). However, the acetate and halide ions have higher affinity for Zn+2 around neutral pH, therefore, we used ZnCl2 at around neutral pH (pH 7.5). Therefore, the Zn+2 is likely to stay in solution, especially at low concentration, as we used 10-12 µM solution. For zinc-protein binding studies, ZnCl2 in HEPES buffer (at pH 7.0 and 8.0) has been used in several studies [33], (Chao and Fu, 2004).
We are grateful to the reviewer for the suggestion as the determination of affinity constants through the ITC titration analysis would further support the reported data. We shall include the ITC analysis to elucidate molecular mechanisms underlying Zn+2 transport by hZnT8 in our future work.
References:
Parsons, D.S.; Hogstrand, C.; Maret, W. The C-terminal cytosolic domain of the human zinc transporter ZnT8 and its diabetes risk variant. FEBS J. 2018, 285, 1237–1250, doi:10.1111/febs.14402.Chao, Y.; Fu, D. Thermodynamic studies of the mechanism of metal binding to the Escherichia coli zinc transporter YiiP. J. Biol. Chem. 2004, 279, 17173–17180, doi:10.1074/jbc.M400208200.
Point 2: (b) The authors use the prediction of structure from CD from other authors (Table 1, reference 54 (reference 33 in the revised version)): why did they not use their own spectra (Fig. 3) (Figure 4 in the revised version) and the Dichroweb page? In that way, the comparison in Table 1 will be more reliable. Moreover, other elements of structure prediction (i.e. random coil and beta-turns) should be included in that table (these other types of “structure” are also predicted (deconvoluted) by FTIR).
Response 2: As per suggestions of the reviewer, Table 1 has been modified. Data from CD spectroscopy recorded during the current study have been added. CD spectra of the WT CTD and the R325W CTD (in the absence of Zn+2, shown in Figure 4) using DichroWeb server at http://dichroweb.cryst.bbk.ac.uk/html/home.shtml [40,41] and the program CONTIN/reference set 3 [42], and using protein concentration independent method (web-based server http://perry.freeshell.org/raussens.html) [39]. The data do not reveal any structural difference between both the CTDs when analyzed using a given technique. Differences in the overall quantitative estimation of the secondary structure content obtained by CD and FT-IR spectroscopic techniques could be due to difference in the technique employed and algorithms used for the analysis of the secondary structure.
Information on other secondary structure elements (i.e. random coil and beta-turns) obtained from FT-IR and CD spectroscopy has been added in Table 1.
References:
Raussens, V.; Ruysschaert, J.-M.; Goormaghtigh, E. Protein concentration is not an absolute prerequisite for the determination of secondary structure from circular dichroism spectra: a new scaling method. Anal. Biochem. 2003, 319, 114–121, doi:10.1016/s0003-2697(03)00285-9. Whitmore, L.; Wallace, B.A. DICHROWEB, an online server for protein secondary structure analyses from circular dichroism spectroscopic data. Nucleic Acids Res. 2004, 32, W668–W673, doi:10.1093/nar/gkh371. Whitmore, L.; Wallace, B.A. Protein secondary structure analyses from circular dichroism spectroscopy: methods and reference databases. Biopolymers 2008, 89, 392–400, doi:10.1002/bip.20853. Provencher, S.W.; Glöckner, J. Estimation of globular protein secondary structure from circular dichroism. Biochemistry 1981, 20, 33–37, doi:10.1021/bi00504a006.
Point 3: (c) Why did the authors not estimate the presence of self-association equilibria from equilibrium ultracentrifugation, instead of using sedimentation velocity (lines 290-295)?
Response 3: It is true that self-association equilibria of the purified protein could be analyzed using equilibrium ultracentrifugation. We have used sedimentation velocity ultracentrifugation in order to confirm the molecular size of the purified protein (~ 40 kDa obtained through gel filtration; Figure 2). The sedimentation velocity method revealed a major peak at ~39.6 kDa, suggesting that the purified protein exists in a tetrameric form (monomeric: ~9.8 kDa) (Figure 2).
Point 4: (d) Further experimental details are required for the CD experiments (i.e., response, scan speed, bandwidth).
Response 4: The following information on the experimental details for the CD experiments has been added:
“Spectra were recorded at a speed of 1 nm/15 sec, sensitivity 50 mdeg, bandwidth 1 nm, resolution 1 nm, and response time 15 sec.” (see 4.7. CD spectroscopy).
Point 5: (e) I think the legend of Fig. S1 is wrong (there are two lanes 4 with different meaning).
Response 5: We are thankful to the reviewer for pointing out this mistake. The legend of Figure S1 has been corrected.
Reviewer 2 Report
The manuscript, "C-terminal domain of the human zinc transporter hZnT8 is structurally indistinguishable from its disease risk variant (R325W)" by Ullah et al describes the expression, purification and biophysical characterization of C-terminal domain of hZnT8. The research work is technically robust, however, it may not add much value to be published for the fact that a similar research work is published in literature. Also this research work is a series of experiments, but not a conclusive story.
Here are few comments for the next revision.
The introduction is too long and disconnected. It would be good if authors reframe the introduction with defining human zinc transporters, its significance in the field and how this research would add the value to the field. A cartoon diagram of hZnT8 to understand different domains and topology of the protein will be helpful for the audience. For figure 2d, curve fitting may be repeated again with another model or equation. It may show up differences between binding affinity of wild type and R325W mutant for Zn and may help in understanding the role of this mutation in diabetes. Authors have quantitatively analyzed secondary structural content using FTIR. Similar information can be obtained by CD spectrum analysis. It would be good to compare information from two sets of experiments. It will be good to emphasize the contribution of these experiments towards understanding the role of R325W mutation in diabetes. In purification, ion exchange step can be deleted and TEV protease can be added before elution and directly be used for reverse Ni-NTA purification. In methods section, procedures for electron microscopy and mass spectrometry are missing.
Round 2
Reviewer 2 Report
Authors have responded to all the suggestions being made. However, introduction could be shortened more.
Author Response
Response to Reviewer 2 Comments
We thank the reviewer for reviewing our manuscript. Please find below our response (highlighted in red) to the reviewer’s comment.
Point 1: Authors have responded to all the suggestions being made. However, introduction could be shortened more.
Response 1: As per suggestion of the worthy reviewer, Introduction has been shortened.